# Predictive Value of HPV, p53, and p16 in the Post-Treatment Evolution of Malignant Tumors of the Oropharynx and Retromolar Trigone–Oropharynx Junction

**DOI:** 10.3390/medicina56100542

**Published:** 2020-10-15

**Authors:** Bogdan Mihail Cobzeanu, Mihail Dan Cobzeanu, Mihaela Moscalu, Octavian Dragos Palade, Luminița Rădulescu, Dragoș Negru, Liliana Gheorghe Moisii, Luiza Maria Cobzeanu, Loredana Beatrice Ungureanu, Patricia Vonica, Daniela Viorelia Matei, Daniela Carmen Rusu, Constantin Volovaț, Victor Vlad Costan

**Affiliations:** Faculty of Medicine, Grigore T. Popa University of Medicine and Pharmacy, 700115 Iasi, Romania; bogdan-mihail.cobzeanu@umfiasi.ro (B.M.C.); octavian.palade@umfiasi.ro (O.D.P.); lmradulescu@yahoo.com (L.R.); draneg@gmail.com (D.N.); moisiil@yahoo.com (L.G.M.); ml_baean@yahoo.com (L.M.C.); lbrungur@yahoo.com (L.B.U.); pvonica@yahoo.com (P.V.); dvm2202@yahoo.com (D.V.M.); danarusu2005@yahoo.com (D.C.R.); cvolovat@gmail.com (C.V.); victorcostan@gmail.com (V.V.C.)

**Keywords:** retromolar trigone–oropharynx junction, HPV, p53, p16, malignant tumors

## Abstract

*Background and objectives:* Knowledge of the interactions and influences of infectious, genetic, and environmental factors on the evolution and treatment response of malignant tumors is essential for improving the management of the disease and increasing patient survival. The objective of this study was to establish the contribution of human papillomavirus (HPV), as well as p53 and p16 tumor markers, alongside associated factors (smoking and alcohol consumption), in the progression of malignancies located in the oropharynx and at the retromolar trigone–oropharyngeal junction. *Materials and Methods:* We performed a prospective study including 50 patients with malignant tumors of the oropharynx and retromolar trigone–oropharyngeal junction. In all patients, the presence and type of HPV were determined, as well as the status of the tumor markers p53 and p16. The associated risk factors, biopsy results, treatment method, and post-treatment evolution were all documented. Statistical analyses were performed to evaluate the correlations between the determining factors and their influence on the post-treatment evolution. An overall increased survival rate was found in HPV(+) patients. *Results:* Our study outlined the prevalence of different high-risk subtypes of HPV from the ones presented by other studies, suggesting a possible geographic variation. Correlations between the p53 and p16 statuses and patient survival could be established. The association of smoking and alcohol consumption strongly correlated with an unfavorable evolution. *Conclusions:* Awareness of the differences in the post-treatment evolution of the patients in relation to the presence of the factors determined in our study could change the future management of such cases for ensuring improved treatment outcomes.

## 1. Introduction

The field literature presents molecular evidence that indicates a correlation between high-risk human papillomavirus (HPV) and the pathogenesis of cancer located in the oropharynx and in the retromolar trigone–oropharyngeal junction region. The expression of HPV with *E6/E7* oncogenes is the current standard criterion for determining the causal role of HPV oncogenes in human tumors. The *E6* and *E7* genes produce oncoproteins that are responsible for p53 degradation, leading to cell-cycle disruption and subsequent cell destruction [1,2].

The modification of p16 by a methylation promoter is frequently observed in squamous cell carcinomas (SCCs) of the head and neck. HPV is readily detectable in HPV(+) tumors and correlates with p16 expression, which is most likely the result of infection with a transcriptionally active HPV [3,4].

The relationship between HPV, p53, and p16 expression in oropharyngeal cancer is known and presented with variable clinical–pathological aspects. Detection using p16 immunohistochemistry can serve, alongside PCR, in the molecular detection of HPV DNA [5].

Oropharyngeal cancer is considered to be a type of mucosal cancer that is associated with HPV infection. HPV is present in 20% of all head and neck cancers and in nearly 60% of tonsillar cancer cases [3,6,7,8].

The expression of p16 protein is a surrogate marker of HPV infection in oropharyngeal cancers. p53 activity mediates cell proliferation in response to mitogenic stimulation; mediates cell-cycle arrest at the level of the G1 phase after DNA damage, thus allowing for repair of the affected DNA before the cell enters the DNA synthesis phase; mediates the induction of apoptosis in cells whose DNA damage is too severe to be repaired. Thus, the inactivation, degradation, or mutation of the p53 gene may disrupt its functions, resulting in cell proliferation, accumulation of damaged DNA, the increased presence of cells carrying DNA errors, and the prolonged survival of the affected cells.

This study aimed to determine the involvement of HPV and p53 and p16 tumor markers, which are associated with factors like alcohol and smoking, in the evolution of malignant oropharyngeal tumors and tumors of the retromolar trigone–oropharyngeal junction.

The presence of these factors influenced the therapeutic behavior of the cases included in the study.

## 2. Materials and Methods

A prospective study was conducted for a period of 3 years (2013–2015) on a batch of 50 patients admitted in the Ear–Nose–Throat (ENT) and Oral and Maxillofacial (OMF) surgery clinics of the Sf. Spiridon County Clinical Emergency Hospital, Iași, Romania, that were diagnosed with malignant tumors located at the oropharyngeal level and at the retromolar trigone–oropharyngeal junction. The main factors involved in the post-treatment evolution were analyzed.

The patients were monitored after treatment until 1 August 2019. The HPV, p53, and p16 tumor markers were dosed in all patients. These aspects were analyzed in the context of the presence of tumor-favoring factors like smoking and alcohol consumption. The anatomopathological features of the biopsy specimens were studied. These were needed to specify the therapeutic conduct, as well as the prognosis and evolution of the cases. Radiological exams—Computed Tomography (CT) scan—were performed in all patients for both diagnostic and post-therapy follow-up, and all cases were discussed by the oncology committee in order to specify the therapeutic approach.

This localization has been shown to be increasing in frequency. The extension of the lesion was confirmed by performing imaging studies and via anatomo-pathology. The involvement of HPV, as well as markers p16 and p53, was investigated using immunological and immunohistochemistry (IHC) studies, all of which influenced the treatment, prognosis, and rate of patient survival.

The study was performed after obtaining informed consent from each patient and the approval of the ethics committee of the Emergency Clinical Hospital Sf Spiridon Iasi, Romania, and of the Grigore T. Popa University of Medicine and Pharmacy of Iasi, Romania (4 March 2015, No. 10477).

The inclusion criteria were age ≥ 18 years and the presence of a malignant tumor located in the oropharynx and at the level of the retromolar trigone–oropharyngeal junction. The exclusion criteria were uncooperative patients, the presence of malignant tumors at other locations, history of oropharyngeal surgery, personal history of primary or secondary motor dysfunction (achalasia, scleroderma peripheral neuropathy, and myopathy), or the presence of clinical signs like unexplained weight loss, hematemesis, hemoptysis, and other general conditions.

### 2.1. Histopathological Exam

The biopsy and surgical excision samples were histopathologically processed to establish the diagnosis.

All specimens were fixed for 18–24 h in 10% neutral buffered formalin at room temperature and then paraffin-embedded. Histological analyses on hematoxylin and eosin (H&E)-stained slides were performed in order to confirm that all blocks contained cancer tissues. Both histopathological and immunohistohistochemical reactions were performed on paraffin-embedded blocks (FFPE). Thirty-one sections of about 4 µm in thickness were obtained from each tissue block according to the HPV-AHEAD protocol. The next steps for processing the sample were deparaffinization, hydration, microwave antigen retrieval treatment, neutralization of endogenous peroxidase, incubation with the primary antibody, and visualization of the antigen–antibody interaction by using the Novolink Polymer Detection System (Leica Microsystems Inc., Newcastle Upon Tyne, UK). As the primary antibody, we used a p53 mouse monoclonal antibody (clone DO-7, Novocastra, Leica Biosystems Newcastle, UK, dilution 1:800, 30 min at 25 °C) and a p16 mouse monoclonal antibody (clone G175-405, BD Pharmingen, dilution 1:25, 60 min at 25 °C). When more than 10% of cells were positive for p53 or p16, the tumor was considered to be positive [9].

The presence of HPV infection was tested by using the HPV-screening kit from AID-Diagnostika (Strassberg, Germany). HPV viral genotyping was performed by using the Opegen^®^ kit ( Zaragoza, Spain). Genomic DNA was isolated from each cytological specimen and amplified via a multiplex PCR technique by using a Seeplex^®^ HPV4A ACE Screening kit (Seegene Inc, Dusseldorf, Germany).

DNA was extracted via an overnight incubation of the paraffin tissue sections in a digestion buffer (10 mM). To minimize the risk of cross-contamination during sectioning, a new blade was used for each FFPE block and the microtome was extensively cleaned after each block with 70% ethanol and DNA Away (Dutscher, Brumath, France). In addition, to monitor possible cross-contamination during the sectioning, empty paraffin blocks were processed every 10th cancer specimen. The HPV DNA positivity was determined by using a type-specific multiplex genotyping (TS-MPG) assay, which combines multiplex PCR and bead-based Luminex technology (Luminex Corporation, Austin, TX, USA) [10].

### 2.2. Statistical Analysis

The statistical data analysis was performed using the SPSS v.25 software (IBM, Armonk, NY, USA). Quantitative variables were reported as an average with its standard deviation. The qualitative variables were presented as absolute and relative frequencies, and the comparisons between groups were made based on the results of the chi-square Pearson or Yates tests. The multivariate analysis of the prediction factors was performed using a logistic regression model. Survival analysis was performed based on the Kaplan–Meier survival curves, which were compared using Cox’s F-test and a log-rank test. The Kaplan–Meier analysis estimates the likelihood of an event (death) of a group over a period. For each time interval, the probability of occurrence of the event was calculated and risk moments were calculated. The significance level calculated in the tests used (*p*-value) was considered significant for *p* < 0.05, where this value represented the maximum accepted probability of error.

The number of patients included in the study was significantly influenced by the fact that in Sf Spiridon County Clinical Emergency Hospital, Iasi, Romania, only a small number of cases (121 cases) were diagnosed with malignant tumors located at the oropharyngeal junction and at the retromolar trigone–oropharyngeal junction during this period. Of these, only those patients who gave their informed consent and met the conditions for inclusion and exclusion were included in the study (50 cases).

## 3. Results

The characteristics of the study group showed a high frequency of male patients (92%), an average age of 47.3 ± 8.6 years, and a frequency of 48% of patients with high school education and 24% with higher education. Alcohol use (48%) and smoking (52%) were important features associated with the studied cases of oropharynx and retromolar trigone–oropharyngeal junction tumors. The coupling of the two aspects (alcohol and tobacco consumption) was an important factor in the development of these tumors, as evidenced by the high percentage of cases (22%) among the included patients (Table 1).

The HPV, p53, and p16 tumor marker analysis revealed p16(+) in 86% of cases, which was a significantly higher frequency (*p* = 0.03) compared to that of the p53(+) identified in 70% of the cases. The frequency of HPV(+) cases (various strains: HPV16, 18, 31, 33, 51, 66) was found in 32% of cases (Table 1). These tumor markers can significantly influence the evolution and staging of the tumor, and thus influence the therapeutic plan for each case [11].

Histopathological findings were specific for HPV involvement (Figure 1). At the same time, in cases with invasive squamous cell carcinoma, the pathological findings emphasized an intense p53 diffuse positivity (Figure 2) that shifted toward an intense p16 positivity for the well-differentiated squamous cell carcinoma (Figure 3).

The most common HPV strain was HPV66 in 14% of the cases (*p* = 0.04). HPV51 was found in 12% of cases; HPV16, 31, and 33 in 4% of cases; HPV18 in 2% of cases. Four cases had associations of two strains: HPV16/51, HPV16/66, HPV51/66, and HPV18/31. A total of 32% of the cases were HPV(+) (16 cases), with 3 cases (6%) having 2 detected HPV strains (HPV16/51, HPV16/66, HPV51/66). All HPV(+) cases were associated with p16 or p16 + p53 tumor markers. The results indicated that out of the HPV(+) cases, 6% (3 cases) were associated with the presence of tumor marker p16 and 26% (13 cases) were associated with the presence of both tumor markers (p53 and p16).

### 3.1. Presence of HPV and Tumor Markers p53 and p16: Therapeutic Approach vs. Evolution

Regarding the association of the HPV presence with the evolution of the cases, this study showed a significant association (χ^2^ = 6.43, *p* = 0.04), which was explained by the favorable evolution of the cases with HPV(+) with a proportion of 81.3% and 0% deaths (Table 2).

The association of a p53 presence with a favorable case evolution was demonstrated on the one hand by the increased frequency of those with a favorable evolution (65.7%) and on the other hand by the significant association (χ^2^ = 9.20, *p* = 0.01) (Table 2). The association of the presence of p16 with the favorable evolution of the cases was evidenced by both the frequency of the cases (62.8%) and the significant association demonstrated in this case (χ^2^ = 14.90, *p* = 0.02) (Table 2).

Cases with HPV(+) had a better prognosis, regardless of the treatment performed, compared to cases with an HPV(−) status, in which relapses and deaths were the most frequent in the case of radiotherapy (83.3%, 15 cases with radiotherapy with relapse or death from 18 resolved cases regardless of treatment and who had a recurrence or death).

An important aspect was the fact that the strains found in our study (HPV31, 33, 55, 66) differed from those usually related to the development of SCCs (HPV16, 18), as found by other studies, thus confirming the geographical specificity.

The results indicated a particular situation that is different from the one presented in the field literature, which can be explained by the specificity of the geographical and territorial area, which had an important impact on the current statistics. The utility of these results will be found in subsequent studies that will evaluate the usefulness of making a vaccine corresponding to the genotype that is specific to the geographical area, as well as establishing the specific therapy for neoplasms with this location. The therapeutic approach showed a significant association with the subsequent evolution of the patients (χ^2^ = 15.74, *p* = 0.015) (Table 3).

There was a significant association between surgery combined with chemo- and radiotherapy and the favorable evolution (75%, *p* = 0.01) (Table 3). Similarly, a significant association was noted for the group of patients undergoing surgery combined with radiotherapy, in which all patients had a favorable evolution (100%, *p* = 0.01) (Table 3). At the same time, the significant association between deaths and the cases that underwent only radiotherapy (28.6%, *p* = 0.03) was noted (Table 3).

### 3.2. Multivariate Analysis on Case Evolution vs. Therapeutic Approach: HPV(+) and the Presence of p53 and p16 Tumor Markers

The study showed that a favorable evolution was significantly associated with p53(+) patients and with the therapeutic approach involving surgery combined with radiotherapy and chemotherapy or surgery combined with radiotherapy (Table 4). This was not surprising, as p53 is associated with invasive squamous cell carcinoma (Figure 2). In the case of patients with p53(−), the favorable evolution was significantly associated with the combination of radiation and chemotherapy or with the combination of surgery and radiotherapy.

In the absence of the p16 marker, a significantly higher rate of deaths (60%) was observed in the patients who underwent radiotherapy. In the case of p16(+), a lower death rate was observed for patients who underwent only radiotherapy (21.8%) (Table 4). These results demonstrated a significant association between the absence of p16 and the increased death rate (χ^2^ = 14.07, *p* = 0.02). Furthermore, for the cases with a favorable evolution, the significant association between the presence of p16 and the therapeutic approach involving surgery combined with radiotherapy and chemotherapy was noted. Recurrence was identified in all cases that utilized surgery combined with radiotherapy and chemotherapy in which p16 was not present.

The results of the multiple regression analysis showed that the presence of p53 (OR = 5.01, 95% CI: 4.51–6.23), the presence of p16 (OR = 4.23, 95% CI: 1.31–4.15), and the presence of HPV(+) (OR = 2.41, 95% CI: 1.74–3.10) could be considered important prognostic (*p* < 0.05) factors regarding the favorable evolution of the cases, significantly increasing the chance of a favorable evolution (Table 5).

### 3.3. The Association of Risk Factors Regarding the Evolution of Cases

In the case of alcohol consumers, a significant number of cases had an HPV(+) status (30%), while patients who did not consume alcohol had an HPV(−) status (38%). A significant proportion of cases that did not consume alcohol and tobacco had the following statuses: HPV(−) (54%), p53(+) (60%), or p16(+) (70%).

In the literature, it is mentioned that the majority of smokers, alcohol consumers, and consumers of both tobacco and alcohol had p53(+) and p16(+) statuses [12]. In our group, 42% of smokers had a p16(+) status and 32% had a p53(+) status, and in the case of alcohol consumers, 32% had a p16(+) status and 30% had a p53(+) status.

Due to the small study group, it was not possible to include all the factors involved in the case evolution in one multivariate analysis. Thus, tobacco and alcohol consumption were introduced in a separate multivariate analysis (Table 6). The results indicated that smoking presented the lowest risk (OR = 1.98, 95% CI: 1.65–3.01, *p* = 0.02) for an unfavorable evolution compared with the risk presented by alcohol consumption (OR = 2.65, 95% CI: 2.06–6.77, *p* < 0.01) or the association of the two favoring factors (OR = 2.82, 95% CI: 2.16–5.88, *p* < 0.01) (Table 6).

### 3.4. Survival Rate Analysis of Patients with Malignant Tumors of the Oropharynx and Retromolar Trigone–Oropharyngeal Junction: Kaplan–Meier Analysis

The prospective study was conducted between 2013 and 2015, with the follow-up of patients who presented at the proposed dates monitored until August 2019 (43 months).

The survival outcome at each moment of the study was evaluated using a life table, and according to the Kaplan–Meier curve graph, the survival rate was estimated (Table 7, Figure 4). There was a significant difference between the survival rates of the two subgroups, depending on the presence/absence of HPV (F = 4.56, *p* < 0.01). The survival rate in the HPV(+) patients reached 80.21% by the end of the study period, which was a significantly higher value compared with the HPV(−) patients, in which the survival rate was 28.12% (Table 7, Figure 4).

Depending on the presence of p53, the Kaplan–Meier analysis of the survival rate showed a slight decrease in both patient groups during the first 15 months post therapy. After this interval, the survival rate decreased significantly in the p53(−) patients, reaching a significant difference between the survival rates at the end of the studied interval (log-rank test: χ^2^ = 2.16, *p* = 0.03) (Table 7, Figure 4). In the p16(+) patients, the Kaplan–Meier analysis indicated an evolution with a higher survival rate, reaching 48.18% at 43 months (log-rank test: χ^2^ = 2.81, *p* < 0.01) (Table 7, Figure 4).

In our study, it was observed that the patients with an HPV(+) status had higher survival rates than the patients with an HPV(−) status. The figure shows that patients with a p53(+) status had a higher survival rate than patients with a p53(−). In patients with a p16(−) status, the survival rate was lower than in patients with a p16(+) status, which corresponds to the existing data from the field literature [13,14].

Most cases with an HPV(−) status had a p16(+) or/and p53(+) status and those with a p53(+) status were also p16(+).

Intraoral access to voluminous tumors of the oral cavity and to most oropharyngeal tumors is often difficult and inadequate.

The sectioning of the mandible offers a good exposure of the entire oral cavity and of the oropharynx. Therefore, it allows for a satisfactory monobloc resection of the primary tumor associated with an appropriate neck dissection.

A paramedian mandibulotomy offers all the advantages of a middle mandibulotomy. Furthermore, it does not disturb the genioglossus, geniohyoid, or digastric muscles, cutting only through the mylohyoid muscle. Thus, the paramedian mandibulotomy would be ideal for access to the oral and oropharyngeal cavity.

Establishing the direct link between HPV infection, in association with other external factors (alcohol, smoking, emotional stress), and oropharyngeal malignancies are important in establishing the most appropriate therapeutic plan for the patient. Determining the exact type of HPV that colonizes the oropharynx in our geographic region could lead to the development of a vaccine similar to that for cervical cancer.

The determination of p53 and p16 tumor markers can improve the prognosis of the patients by selecting the best treatment approach.

An early diagnosis, along with implementing prophylactic measures (adequate hygiene, counseling against smoking and alcohol consumption, sexual education) can improve the survival rate and quality of life for these patients and also reduce treatment costs [12]. Our results demonstrated that:HPV(+) patients had a longer survival time than HPV(−) patients.Patients with a p53(+) status had a longer survival time than p53(−) patients.Patients with a p16(+) status had a longer survival time than p16(−) patients.

## 4. Discussion

The involvement of human papillomavirus (HPV), as well as p53 and p16 biomarkers, have been shown to be involved in the genesis of cancers located in the oropharynx, as well as the junction of the retromolar trigone with the oropharynx, a particular area due to the fact that it is at the border of two territories, namely, ENT and OMF surgery.

We aimed to determine the exact HPV subtype for the cases in which the virus was detected.

Standard investigations were performed, including histopathological analysis of the biopsies, CT, MRI, and ultrasound examinations, in order to specify the topographic limits of the lesions, taking into account the anatomical particularities of the region and the proximity of important vascular and nervous landmarks, while aiming for the precise evaluation of the local extension.

The number of patients with tumors arising in the studied anatomical region was too low to have a major statistical significance but it was similar to the numbers reported by other specialized studies. Of the 50 patients tested so far, 16 had a high-risk HPV infection, with this parameter being very important in the evaluation of the therapeutic approach and the prognosis.

Due to the rarity of the condition located at the level of the retromolar trigone–oropharyngeal junction, the number of patients was low, which prevented us from performing a complex statistical analysis. However, the post-therapeutic results were superior for the cases detected with high-risk HPV.

The presence of the subtypes 16 and 18, described as the most frequent by the field literature, was not detected in our study, which suggested a geographical distribution of the HPV subtypes, hence the need to obtain personalized vaccines for each geographical region. In patients with high-risk HPV, the prognosis after surgery was better than in HPV(−) cases or with other phenotypes.

The post-therapeutic results were better in the cases in which a high-risk HPV infection was determined. Smoking and chronic alcohol consumption are traditionally accepted risk factors for SCCs of the head and neck, to which HPV infection has been added as a high-risk factor that is associated with certain subsets of head and neck neoplasms.

Patients with HPV(+) neoplasms tend to be younger than those with HPV(−), probably due to differences in sexual behavior [15,16,17].

The most common HPV subtype detected according to the literature is HPV16, representing approximately 90% of all HPV(+) types involved in head and neck SCCs [18,19].

The involvement of HPV in oropharyngeal neoplasms has shown a favorable treatment response and a good prognosis with the expression of p16, which may be considered a marker for the identification of HPV(+) neoplasms in the head and neck region [18,19].

High-molecular-expression p53 mutations are involved in neoplasms found in chronic smokers [5,20].

The association between HPV status and a vital prognosis was proven, with HPV(+) patients demonstrating a higher survival rate than HPV(−) patients [18,19,21].

The tests used to detect HPV were in situ hybridization; immunohistochemistry with HPV, p53, and p16 as markers; PCR. HPV is responsible for a variety of skin and mucosal diseases. Clinical manifestations may vary depending on the anatomical location of the lesion and the type of HPV involved. HPV strains are generally divided into two categories: those with a potential low oncogenic risk (low-risk group) and those with a potential medium–high oncogenic risk (high-risk group). High-risk HPV strains are generally associated with precancerous lesions and invasive cancer, while low-risk HPV strains are commonly found in asymptomatic or benign conditions [3,7,16,22].

High-risk HPV (HR-HPV) comprises types 16, 18, 31, 33, 35, 39, 45, 51, 52, 56, 58, 59, 68, 69, 73, and 82.

Low-risk HPV (LR-HPV) comprises types 6, 11, 40, 42, 43, 44, 54, 61, 70, 72, 81, and 89.

HPV with an intermediate risk comprises types 26, 53, and 66.

HPV with indefinite risk comprises types 34, 55, 57, and 83.

The presence of the subtypes 16 and 18, with the highest frequency in the field literature, was not detected in our study, where the most frequent ones were instead the 31, 33, 55, and 66 subtypes, which highlighted a geographical distribution of HPV subtypes [3,7,23].

Several studies have shown that patients with HPV(+) oropharyngeal cancer, as identified using PCR, in situ hybridization, or immunohistochemistry for p16 in tumor tissue, have significantly improved overall survival compared with patients with HPV(−) oropharyngeal cancer [2,23,24].

HPV(+) tumors may have fewer genetic changes that may be associated with a better response to therapy. HPV(+) tumors have higher radiosensitivity, probably due to the intact apoptotic response to radiation [25].

The immunological response may play a role in improving the response to radiation and chemotherapy in HPV(+) tumors (due to the stimulation of the direct immune response to specific viral tumor antigens) [26].

The younger age, better overall condition, and fewer comorbidities of HPV(+) patients with oropharyngeal cancer may contribute to an improved survival rate [27].

All these data suggest that HPV status can be used in decision-making to select patients for a less aggressive treatment than surgical treatment. p16 immunohistochemistry (IHC) is a current marker used to detect the presence of HPV.

The biology of HPV(+) oropharyngeal cancer is characterized by p53 degradation, inactivation of the RB pathway by retinoblastoma, and the up-regulation of p16. In contrast, tobacco-associated oropharyngeal cancer is characterized by the mutation of p53 and down-regulation of p16. HPV(+) oropharyngeal cancer appears to be more receptive to chemotherapy and radiation than HPV(−) [2].

Given the HPV(+) and HPV(−) oropharyngeal cancers and their molecular changes in p16 and p53 gene expression, we can conclude that patients with HPV(+) cancer with a higher expression of p16 and a lower expression of p53 respond better to treatment and have a better prognosis. The T stage in the TNM (TNM Classification of Malignant Tumors—Tumor, Nodes, Metastasis) classification differs depending on the HPV status. HPV(+) tumors are more likely to have a lower T stage than HPV(−) tumors, with an advanced T stage being a significant risk factor for relapse and death. Patients with a p16(+) status in oropharyngeal tumors are less likely to have persistent lymph node metastases in the cervical lymph nodes after chemotherapy and it is thus hypothesized that neck dissection may be avoided [8,28]. Several authors indicate that the impact of the HPV status needs to be evaluated through prospective clinical trials. The current staging system of the American Cancerology Committee (AJCC) for oropharyngeal cancer should be modified to better reflect the prognosis in patients with vegetative or infiltrative HPV(+) or HPV(−) tumors [17,29,30].

HPV(+) tumors represent a new well-defined entity, which may develop even at an early age in individuals with a good performance status, especially males, that can be non-smokers and non-alcoholics and could have different sexual behaviors [6,29,31].

Oncologists need to be aware of these new findings and HPV PCR testing should be performed routinely since the results have a significant impact on the prognosis, which influences the therapeutic approach. However, changes occur fast in this field and we expect the HPV evaluation will be readily available for these patients in the near future.

The HPV genome has roughly 8000 bp (base pairs) and encodes 8 genes whose expression is closely related to the differentiation process, thus making it possible to complete the life cycle of the human papillomavirus in any multi-layered epithelial tissue. The main target cells of the HPV infection are the basal cells of epithelial tissues. After the infection of the cells from the basal layer, the viral genome is maintained in the form of 50–100 extrachromosomal circular elements (nuclear episomes) per infected cell. The first expressed viral genes are *E6* and *E7*. These signal the entry of the infected cell into the cell cycle and activation of DNA polymerase for the replication of both viral DNA and host DNA. During the process of differentiation of the basal cell and its ascension from the germinal layer into the distal layers, there is a decrease in the importance of the expression of *E6* and *E7* genes and the activation of a secondary promoter that leads to the transcription of primary genes (*E1, E2, E4*, and *E5*). These primary genes initiate amplification of the viral genome in over 1000 copies per cell.

Thus, when the cell reaches the final stage of differentiation, the viral capsid proteins (L1 and L2) are expressed and assembly of the resulting virions takes place. These virions are released with the death of the host cell [25,32,33].

The carcinogenic activity of high-risk human papillomavirus types is largely due to the expression of the two initial viral genes (*E6* and *E7*). These “oncogenes” degrade and destabilize p53 and retinoblastoma 1 (pRB), which are two important oncosuppressor proteins. *E6* protein leads to the formation of a complex between p53 and *E3* ubiquitin ligase. Thus, ubiquitination of the p53 protein and its degradation by the cellular proteasome takes place.

Decreased p53 cell concentration results in both loss of control of the G1/S and G2/M transition of the cell cycle and increased cellular stress, with serious repercussions on the stability of the infected cell genome. Binding of the *E7* viral protein to pRb induces degradation of the latter via recruitment of the cullin 2 ubiquitin ligase complex. These two actions on pRb (*E7*-pRb interaction and pRb degradation via the proteasome) lead to the expression of the transcriptional factors in the *E2F* family involved in activating genes involved in the S-phase transition of the cell cycle, allowing for uncontrolled proliferation of the infected cells.

The proteins encoded by the viral genes *E6* and *E7* not only interfere with p53 and pRb activity but also with other proteins involved in the cell cycle control or signaling cascades required for programmed cell proliferation, differentiation, and death.

Of major importance in the HPV-induced tumor transformation process is the interaction between protein *E6* and PDZ proteins involved in maintaining intracellular connections and apical–basal polarization. Thus, the transformation of infected epithelial cells into mesenchymal cells takes place, which is an extraordinarily important process for inducing the metastasis capacity of tumor-transformed cells.

HPV(+) cancers are associated with very good survival, despite advanced tumor stages. In the case of squamous cell oropharyngeal tumors, determining the HPV status at the beginning of the study using the immunohistochemical expression of p16INK4a together with the detection of the HPV DNA using PCR can be used as prognostic indicators. Further studies are needed to determine the risk factors for HPV-induced tumors [21,24,34].

The *TP53* gene has the role of promoting the secretion of the p53 protein, which is a tumor suppressor that controls and regulates the cell cycle, thus preventing premature cell division or abnormal division. The p53 protein located in the nucleus directly binds to DNA. In the case where the DNA is damaged due to various causes (toxic substances, environmental factors, pollution, UV radiation, etc.), the p53 protein determines whether the DNA can be repaired or whether the cell will commit suicide through the phenomenon of apoptosis, thus preventing the appearance of tumors. If the DNA can be repaired, p53 will train other genes to participate in the repair process; if it cannot be repaired, it gives a signal that will induce apoptosis of the damaged DNA cell [35].

The *TP53* gene is located on the short arm of chromosome 17 at position 13.1. Somatic mutations of the *TP53* gene were found in approximately 50% of head and neck SCCs. Most genetic mutations of the *TP53* gene involved in head and neck carcinomas consist only in changing an amino acid, which leads to protein dysfunction and therefore its inability to exert control over cell division; in this case, damaged DNA accumulates in the cell that continues to divide chaotically, leading to the appearance of the tumor [36].

In conclusion, a new TNM staging for HPV(+) tumors that includes a specific HPV gene and antiviral therapy is desired.

A limitation of the study was the small number of included patients since the location of the malignant condition in the retromolar trigone–oropharyngeal junction is quite rarely encountered, thus preventing a complex statistical analysis. At the same time, the anatomical position of the retromolar trigon area made the therapeutical approach challenging due to it being on the border between oral and maxilo-facial surgery and otorhinolaryngology.

## 5. Conclusions

The most positive prognostic factors were the positivity of p53, p16, and HPV, while the most common negative factor was the combined consumption of alcohol and tobacco.

It was not surprising that there was an important association between HPV(+) and neoplasms. Our study indicated a particular situation that is different from the one presented in the field literature, which can be explained by the specificity of the geographical and territorial area, which had an important impact on the current statistics. The utility of these results will be found in subsequent studies that will evaluate the usefulness of making a vaccine corresponding to the genotype that is specific to the geographical area, as well as establishing a specific therapy for neoplasms with this location.

These tumors have a continuous rise in frequency in non-smokers and non-chronic alcohol consumers. The mean age for diagnosis is lowering due to changes in sexual behavior.

An adaptation of the TNM staging is desired considering that the HPV(+) or HPV(−) statuses of a patient’s tumor permit a better development of therapeutical protocols into the oncological boards.

## Figures and Tables

**Figure 1 medicina-56-00542-f001:**
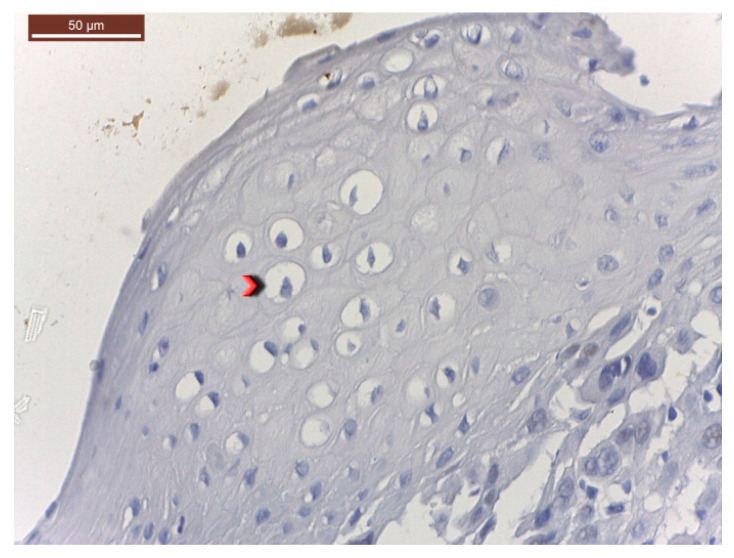
Surface squamous epithelium with HPV-type cytopathic changes (koilocytes—red arrowhead; HE, ×400).

**Figure 2 medicina-56-00542-f002:**
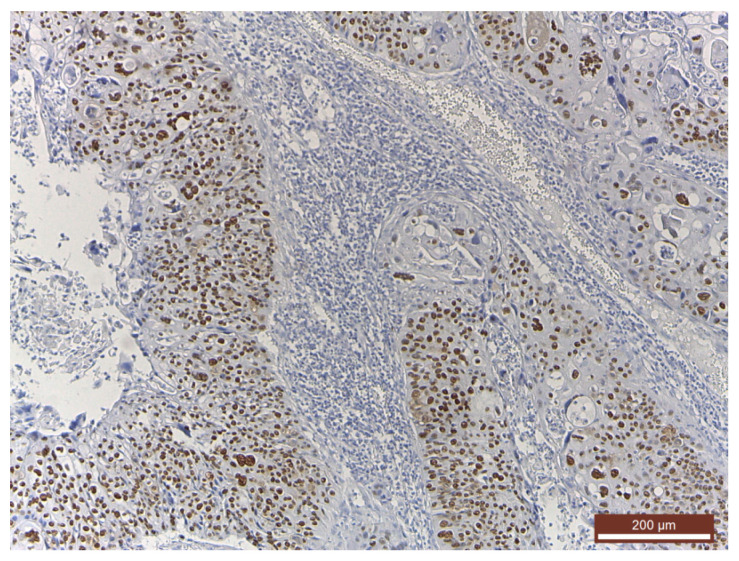
Intense diffuse p53 positivity in an invasive squamous cell carcinoma (immunohistochemistry (IHC), anti-p53 Ab, ×100).

**Figure 3 medicina-56-00542-f003:**
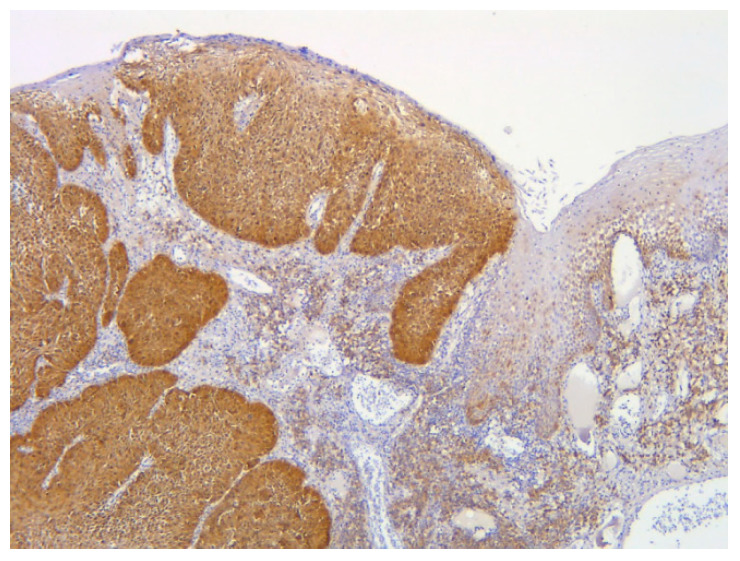
Diffuse intense p16 positivity in an invasive well-differentiated squamous cell carcinoma and a suprajacent squamous epithelium (IHC, anti-p16 Ab, ×100).

**Figure 4 medicina-56-00542-f004:**
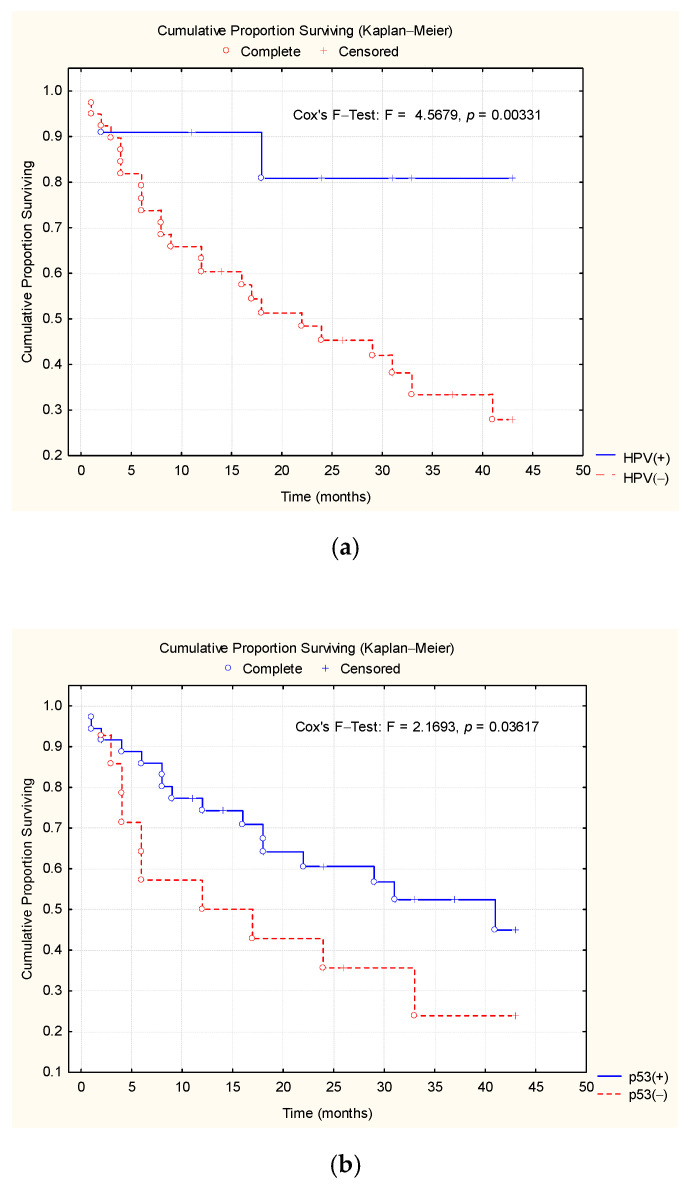
The Kaplan–Meier survival curves. Patients’ survival rates were estimated based on the presence or absence of (**a**) HPV, (**b**) p53, and (**c**) p16.

**Table 1 medicina-56-00542-t001:** The clinical and sociodemographic characteristics of the patients with malignant tumors of the oropharynx and the retromolar trigone–oropharynx junction.

Characteristics	Study Group (*n* = 50)	*p*-Value ‡
Age (years), (mean ± SD)	47.3 ± 8.6	
Male/Female, *n* (%)	46 (92%)/4 (8%)	<0.01 *
Environment Rural/Urban, *n* (%)	21 (42%)/29 (58%)	0.04 *
Elementary school, *n* (%)	14 (28%)	0.02 *
High school, *n* (%)	24 (48%)	
Higher education, *n* (%)	12 (24%)	
Tobacco consumption, Yes/No, *n* (%)	24 (48%)/26 (52%)	0.03 *
Alcohol consumption, Yes/No, *n* (%)	21 (42%)/29 (58%)	0.03 *
Alcohol and tobacco consumption, Yes/No, *n* (%)	11 (22%)/39 (78%)	0.01 *
HPV (+), *n* (%)	16 (32%)	0.03 *
p53 present, *n* (%)	35 (70%)	
p16 present, *n* (%)	43 (86%)	
HPV18 (+), *n* (%)	1 (2%)	0.04 *
HPV33 (+), *n* (%)	2 (4%)	
HPV31 (+), *n* (%)	2 (4%)	
HPV16 (+), *n* (%)	2 (4%)	
HPV51 (+), *n* (%)	6 (12%)	
HPV66 (+), *n* (%)	7 (14%)	
Radiotherapy and chemotherapy, *n* (%)	12 (24%)	0.01 *
Surgery associated with radiotherapy and chemotherapy, *n* (%)	4 (8%)	
Surgery and radiotherapy, *n* (%)	6 (12%)	
Radiotherapy, *n* (%)	28 (56%)	

‡ Pearson chi-square or Yates test; * marked effects were significant at *p* < 0.05. HPV: human papillomavirus.

**Table 2 medicina-56-00542-t002:** Association of HPV presence with p53 and p16 vs. the evolution of cases with the malignant tumors of the oropharynx and the retromolar trigone–oropharynx junction.

HPV, p53, or p16 (*n* = 50)	Evolution, *n* (%)			*p*-Value ‡
	Favorable	Relapse	Death	
HPV present/	13 (81.3%)/	3 (18.8%)/	−/	0.04 *
absent	16 (47.1%)	10 (29.4%)	8 (23.5%)	
p53 present/	23 (65.7%)/	10 (28.6%)/	2 (5.7%)/	0.01 *
absent	6 (40%)	3 (20%)	6 (40%)	
p16 present/	27 (62.8%)/	11 (25.6%)/	5 (11.6%)/	0.02 *
absent	2 (28.6%)	2 (28.6%)	3 (42.9%)	

‡ Pearson chi-square test; * marked effects were significant at *p* < 0.05.

**Table 3 medicina-56-00542-t003:** Association of the type of therapy vs. the evolution of cases with the malignant tumors of the oropharynx and the retromolar trigone–oropharynx junction.

Type of Therapy (*n* = 50)	Evolution, *n* (%)			*p*-Value ‡
	Favorable	Relapse	Death	
Surgery combined with radiotherapy and chemotherapy	3 (75%)	1 (25%)	−	0.01 *
Radiotherapy and chemotherapy	8 (66.7%)	4 (33.3%)	−	0.02 *
Surgery and radiotherapy	6 (100%)	−	−	<0.001 *
Radiotherapy	12 (42.9%)	8 (28.6%)	8 (28.6%)	0.03 *
All groups				0.015 *

‡ Pearson chi-square test; * marked effects were significant at *p* < 0.05.

**Table 4 medicina-56-00542-t004:** Association of the case evolution with malignant tumors of the oropharynx and the retromolar trigone–oropharynx junction with HPV(+), with the presence of tumor markers p53 and p16 and the type of therapy.

HPV (+), p53, and p16	Type of Therapy	Evolution, *n* (%)			*p*-Value ‡
		Favorable	Relapse	Death	
HPV positive	Surgery combined with radiotherapy and chemotherapy	2 (100%)	−	−	0.01566 *
	Radiotherapy and chemotherapy	2 (50%)	2 (50%)	−	
	Surgery and radiotherapy	5 (100%)	−	−	
	Radiotherapy	4 (80%)	1 (20%)	−	
p53 present	Surgery combined with radiotherapy and chemotherapy	3 (100%)	−	−	0.01411 *
	Radiotherapy and chemotherapy	6 (60%)	4 (40%)	−	
	Surgery and radiotherapy	4 (100%)	−	−	
	Radiotherapy	10 (55.6%)	6 (33.3%)	2 (11.1%)	
p16 present	Surgery combined with radiotherapy and chemotherapy	3 (100%)	−	−	0.02879 *
	Radiotherapy and chemotherapy	7 (63.6%)	4 (36.4%)	−	
	Surgery and radiotherapy	6 (100%)	−	−	
	Radiotherapy	11 (47.8%)	7 (30.4%)	5 (21.8%)	

‡ Pearson chi-square test; * marked effects were significant at *p* < 0.05.

**Table 5 medicina-56-00542-t005:** Multiple regression analysis results. Evaluation of the degree of prediction according to HPV(+) and the presence of tumor markers p53 and p16 on the evolution of cases with malignant tumors of the oropharynx and the retromolar trigone–oropharynx junction.

Multiple Regression	β	SE	Wald	*p*-Value	Odds Ratio (OR) Exp(β)	95% CI for Exp(β)	
						Lower	Upper
p53 (+)	5.87	0.25	20.07	<0.01 *	5.01	4.51	6.23
p16 (+)	4.03	0.19	2.97	0.03 *	4.23	1.31	4.15
HPV (+)	2.09	0.24	3.95	0.01 *	2.41	1.74	3.10

CI—confidence interval, SE—standard error; Hosmer and Lemeshow test: χ^2^ = 10.21, *p* = 0.157; R^2^ = 0.684; standard error of the estimate = 0.0268; * marked effects were significant at *p* < 0.05.

**Table 6 medicina-56-00542-t006:** Multiple regression analysis results. Evaluation of the factors involved in the evolution of cases with malignant tumors of the oropharynx and the retromolar trigone–oropharynx junction.

Multiple Regression	β	SE	Wald	*p*-Value	Odds Ratio Exp(β)	95% CI for Exp(β)	
						Lower	Upper
Tobacco consumption	2.96	0.22	8.70	0.02 *	1.98	1.65	3.01
Alcohol consumption	5.01	0.21	11.52	<0.01 *	2.65	2.06	6.77
Alcohol and tobacco consumption	6.97	0.31	13.87	<0.01 *	2.82	2.16	5.88

CI—confidence interval, SE—standard error; Hosmer and Lemeshow test: χ^2^ = 4.67, *p* = 0.0989; R^2^ = 0.6791; standard error of the estimate = 0.0187; * marked effects were significant at *p* < 0.05.

**Table 7 medicina-56-00542-t007:** Lifetable for the evaluation of the post-therapy survival rate in the study group.

Lower Limit for Time (Months)	Cumulative Proportion Surviving					
	HPV(+)	HPV(−)	p53(+)	p53(−)	p16(+)	p16(−)
1.0	100	100	100	100	100	100
5.6	90.90	81.81	88.73	71.42	91.54	64.28
10.3	90.90	65.72	77.09	57.14	77.01	57.14
15.0	90.90	60.01	73.94	50.00	70.73	57.14
19.6	80.21	51.01	63.63	42.85	63.83	42.85
24.3	80.21	45.01	59.99	35.06	60.07	35.71
29.0	80.21	45.01	59.99	35.06	60.07	35.71
33.6	80.21	33.75	50.76	23.37	55.06	17.85
38.3	80.21	33.75	50.76	23.37	55.06	17.85
43.0	80.21	28.12	43.51	23.37	48.18	17.85

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
