# Peer review of "Predictive Value of HPV, p53, and p16 in the Post-Treatment Evolution of Malignant Tumors of the Oropharynx and Retromolar Trigone–Oropharynx Junction"

_medicina, 2020, doi:10.3390/medicina56100542_

Round 1
Reviewer 1 Report
This is a logical analysis of patient cohort with regards to HPV status, type and p53 and pRb status.
Although a small number, they provide statistical power to show that HPV positive, p53 positive and pRb positive patients survival rate is higher.
It is interesting that the most well-studied high-risk types (16 and 18) were not detected in this study - something that may be worth investigating further in the future, perhaps with a greater number of patient samples (I am not suggesting that this is within the scope of the current study).
Altogether well-written and presented. I have no further suggestions or comments.
Author Response
A limitation of the study is the small number of included patients since the location of the malignant condition in the retromolar trigone-oropharyngeal junction is quite rarely encountered, thus preventing a complex statistical analysis. In the same time, anatomical position of the retromolar trigon area made the therapeutical approach challenging between oral and maxilo-facial surgeon and otorhinolaryngology.
Reviewer 2 Report
Review –
The authors took up a very important problem from a clinical point of view. The aim of this study was to determine the involvement of HPV and p53 and p16 tumor markers, associated with factors like alcohol and smoking, in the evolution of malignant oropharyngeal cancers.
The results were described in detail and presented graphically and in tables. The statistical analysis was performed correctly.
I would emphasize the speculations of your results and explain why your finding would be beneficial to clinical practice.
My comments are following:
Major points:
1. The materials and methods used in the research are not clearly described. Materials and methods should be corrected. The authors did not write about the clinical material in which DNA was detected – fresh frozen tissue or paraffin embedded.
- What methods and what diagnostic tests were used for detection HPV DNA and HPV genotyping. What method was used to determine the p53 and p16 tumor markers. The source of this method should be given.
- Are there any limitations to this research?
- The discussion needs to be corrected
Line 168-186 – please move to the discussion
Line 294-297 as above
The text will then be clearer to the reader.
- There are no conclusions from the conducted research. What application in clinical practice may have these results? Conclusions may not be a repetition of the results.
Some minor comments:
- Line 67 ENT OMF - abbreviation must be explained because it appears for the first time;
- The literature should be supplemented with newer items.
Author Response
Major points:
Answer for points 1 and 2.
We reformulated section 2.1. Histopathological exam.
I have kept lines 92 and 93 in this section but replaced lines 94-101 (in the original manuscript) with other paragraphs.
All specimens were fixed for 18–24 hours in 10% neutral buffered formalin, at room temperature and then paraffin embedded. Histological analyses on hematoxylin and eosin (H&E) stained slides were performed in order to confirm that all blocks contain cancer tissues. Both histopathological and immunohistohistochemical reactions were performed on paraffin embedded blocks (FFPE). 31 sections of about 4 µm thickness were obtained from each tissue block according to the HPV-AHEAD protocol. The next steps for processing the sample were deparaffinization, hydration, microwave antigen retrieval treatment, neutralization of endogenous peroxidase and incubation with the primary antibody, visualization of antigen-antibody interaction by using Novolink Polymer Detection System. As primary antibody we used p53 mouse monoclonal antibody (clone DO-7, Novocastra, Leica Biosystems Newcastle, UK, dilution 1: 800, 30 minutes at 25 °C) and p16 mouse monoclonal antibody (clone G175-405, BD Pharmingen, dilution 1:25, 60 min. at 25 °C). When more than 10% cells were positive for p53 or p16, the tumor was considered to be positive [9].
The presence of HPV infection was tested by using the HPV-screening kit from AID-Diagnostika, Germany. HPV viral genotyping was performed by using the Opegen® kit (Spain). Genomic DNA was isolated from cytological specimen and amplified by multiplex PCR technique by using Seeplex® HPV4A ACE Screening kit (Seegene Inc,Germany).
DNA was extracted by an overnight incubation of the paraffin tissue sections in a digestion buffer (10 mM). To minimize the risk of cross-contamination during sectioning, a new blade was used for each FFPE block and the microtome was extensively cleaned after each block with ethanol 70% and DNA Away (Dutscher, Brumath, France). In addition, to monitor possible cross-contamination during the sectioning, empty paraffin blocks were processed every 10th cancer specimen. HPV DNA positivity was determined by using a type-specific multiplex genotyping (TS-MPG) assay, which combines multiplex PCR and bead-based Luminex technology (Luminex Corporation, Austin, TX) [10].
For these new paragraphs we have added two new bibliographic references [9] and [10].
9. Ando K.; Oki E.; Saeki H.;, Yan Z.; Tsuda Y.; Hidaka G., Kasagi Y.; Otsu H.; Kawano H.; Kitao H.; Morita M.; Maehara Y. Discrimination of p53 immunohistochemistry-positive tumors by its staining pattern in gastric cancer. Cancer Med. 2015, 4(1), 75-83. doi: 10.1002/cam4.346.
10. Gheit T.; Abedi-Ardekani B.; Carreira C.; Missad C.G.; Tommasino M.; Torrente M.C. Comprehensive analysis of HPV expression in laryngeal squamous cell carcinoma. J Med Virol. 2014, 86(4), 642–646, https://doi.org/10.1002/jmv.23866.
Thus, we renumbered the initial references existing in the manuscript, starting with the reference [9].
Answer to point 3.
We defined the limitations of the research.
I mentioned (introduced) the limitations of the research after line 438 (corresponding to the initial manuscript) - before section 5. Conclusions.
A limitation of the study is the small number of included patients since the location of the malignant condition in the retromolar trigone-oropharyngeal junction is quite rarely encountered, thus preventing a complex statistical analysis. In the same time, anatomical position of the retromolar trigon area made the therapeutical approach challenging between oral and maxilo-facial surgeon and otorhinolaryngology.
Answer to point 4.
I moved lines 294-303 (corresponding to the original manuscript) which I inserted in the discussion section starting from line 316 (corresponding to the initial manuscript).
Answer to point 5.
We reformulated all the conclusions according to the recommendations.
5. Conclusions
The most positive prognostic factors are the positivity of p53, p16 and HPV while the most common negative factor is combined consumption of alcohol and tobacco.
It is not surprising that there is an important association between HPV(+) and neplasms. Our study indicates a particular situation that is different from the one presented in the field literature, which can be explained by the specificity of the geographical and territorial area with important impact on the current statistics. The utility of these results will be found in subsequent studies that will evaluate the usefulness of making a vaccine corresponding to the genotype specific to the geographical area, as well as establishing the specific therapy for neoplasms with this location.
These tumors have a continuous rise in frequency, in non-smokers and non-chronic alcohol consumers. The mean-age for diagnosis is lowering due to changes in sexual behaviour.
An adaptation of TNM staging is desired considering the HPV(+) or HPV(‒) status of the patient's tumor to permit a better approaching of therapeutical protocols into the oncological boards.
Some minor comments:
Answer to point 1.
In line 67 I explained the abbreviation ENT and OMF:
Ear-Nose-Throat (ENT) and Oral and Maxillofacial (OMF)
Answer to point 2.
Regarding the year of publication of the bibliographic references, we mention the fact that we selected from all the analyzed references those references that made strict reference to the aspects studied by the authors of this study. We have introduced new references corresponding to section 2.1. Histopathological exam
